# Segmenting Carotid Plaques with Different SAM Prompts

**Zhiwei Zhang**[1]                                        ZZHANG679@WISC.EDU

**Yimeng Dou**[1]                                           YDOU8@WISC.EDU

**Robert J Dempsey**[2]                    DEMPSEY@NEUROSURGERY.WISC.EDU

**Carol C. Mitchell**[3]                          CCM@MEDICINE.WISC.EDU

**Tomy Varghese** [1]                              TVARGHESE@WISC.EDU

[1] *University of Wisconsin-Madison, Department of Electrical and Computer Engineering*

[2] *University of Wisconsin-Madison, Department of Neurological Surgery*

[3] *University of Wisconsin-Madison, Department of Medicine*

**Editors:** Accepted for publication at MIDL 2025

## Abstract

Large foundation models have been built to address B-mode ultrasound segmentation for regions with homogeneous morphology. However, most of the foundation models still rely on human input to provide interactive prompting during inference. We explored the prompt type and generation when utilizing the ultrasound foundation model, UltraSAM for the plaque segmentation problem and compared the prompt performance with the ground truth baselines. The results showed that automatic generated prompt variants can reach similar performance with ground truth prompting in the bounding box segmentation of ultrasound plaque segmentation.

**Keywords:** Ultrasound Imaging, foundation model, deep learning, image segmentation

## 1. Introduction

Atherosclerosis is one of the most common systemic vascular diseases(Stefanadis et al., 2017) and according to the Center for Disease Control and Prevention (CDC), cerebrovascular and cardiovascular diseases are the leading cause for both death and disability in the entire world. Studies have shown that the intima-media thickness (IMT) of the carotid artery can be used as a predictor of stroke (xiao Chen et al., 2023), indicating that carotid wall and plaque segmentation being a key factor to evaluate early risk reduction and identify plaque stability via ultrasound imaging (Kiernan et al., 2024). Thus, modern approaches are essential for efficient and accurate carotid plaque segmentation and various approaches have been proposed including utilizing deep learning methods and foundation models since these models has been applied vastly in natural image segmentation tasks. Kirillov et al. (2023) proposed the Segment Anything Model (SAM) which uses an instance specific prompt, such as points, bounding boxes and masks to generate segmentation masks from information embedded in the provided prompt. Zhu et al. (2024) finetuned the SAM model on multiple medical imaging modalities and proposed MedSAM and MedSAM2 model for both 2D and 3D cases. Specifically for ultrasound image segmentation tasks, Meyer et al. (2024) finetuned the SAM model on a utilized ultrasound dataset known as US-43d which is a collection of 43 datasets from different clinical applications. However, both SAM and UltraSAM still relies on the human input when generating the prompt, where the point and box prompts used during testing and inference are simulated from selected ground truth masks as the

human interactive prompts by choosing the center point from the ground truth mask as point prompt and calculating the box from the ground truth mask as box prompt. In contrast, it is difficult to include and require prior knowledge for plaque segmentation and ideally due to complex structure and heterogeneity, it will be more efficient to automate the prompt generalize process without using human resources. In this paper, we explore several methods to generate prompt for foundation model applications in carotid plaque segmentation. We chose to use UltraSAM as the base model, including using a MaskRCNN network to generate a box prompt, noted as MaskRCNN-UltraSAM, and using the first frame of a video sequence to introduce grounding prompts to the foundation model, noted as MFG-UltraSAM. The results of these methods show that sufficient prompting can improve the performance of the foundation model and could pave the direction for future research.

## 2. Methods and Experiments

### 2.1. MaskRCNN generated prompt

In the literature, multiple approaches has been used to segment ultrasound images based on MaskRCNN and U-net networks. Kiernan et al. (2024) has shown that by training a MaskRCNN network, carotid lumen areas can be segmented from single-channel B-mode training inputs and the bounding box predicted by the MaskRCNN has shown high performance. Similarly, by training a MaskRCNN network, it can also predict the bounding box for plaque areas as well. The MaskRCNN model utilizes a ResNet50 backbone and uses both region proposal network (RPN) and ROI pooling to detect the final bounding boxes. After predicting the bounding box area, we transformed the predicted xy-coordinate, height and weight information into a Common Objects in Context (COCO) format prompt, and then these are used as input as well as the original image during inference.

### 2.2. Multi frame grounding prompt

In case of clinical use, it is more efficient for sonographers to create a prompt for one frame compared to segmenting every frame in the sequence. This concept is also similar as grounding or Chain-of-Thoughts in large language models for using case specific information without training knowledge (Wei et al., 2023). Thus, when dealing with multi frame ultrasound image sequences, the prompt can be locally utilized by using the first frame in the sequence which serves as the intermediate steps for the following frames to arrive to the final segmentation. In this experiment, we compare the results for using different prompts for each frame individually and using the previous predicted mask as prior information to generate the point prompt for the next frame in the sequence.

### 2.3. Experiment Details

The plaque image dataset we used consists of B-Mode images which are converted from RF data loops that were acquired by keeping the transducer relatively stationary on the subject's neck. Meshram et al. (2017) has documented this dataset in previous literature reports for Lagrangian strain imaging. A total of 129 images from 43 different sequences were randomly selected to form the testing dataset whereas the remaining form the training and validation dataset for finetuning UltraSAM with a 70-15-15 split. This version is noted

as the zeroshot-UltraSAM. To evaluate how the network performs for the segmentation task, average precision (AP) and average recall (AR) for different IoU confidence levels(0.5 to 0.95), as well as selected confidence level mAP50 and mAP75 are calculated and quantified.

## 2.4. Results

Table 1 and Table 2 show the results of using different prompts when utilizing the Ultra-SAM model. As shown in the table, using MaskRCNN generated prompt and the multi frame grounding prompt both reach comparable results when compared to that of using the prompt with the ground truth in terms of bounding box segmentation. The MaskRCNN-UltraSAM reaches the highest value of 98% in mAP50. In terms of plaque area mask segmentation, the MFG-UltraSAM is the closest in comparison to UltraSAM results, which shows that both methods can reach similar results of the baseline without the ground truth segmentation available during prompting.

Table 1: Bounding Box Segmentation Quantification Results

| Model | prompt type | mAP | mAP50 | mAP75 | mAR |
|---|---|---|---|---|---|
| UltraSAM | point | 74.7 | 90.1 | 88.9 | 78.2 |
| UltraSAM-zeroshot | point | 38.6 | 86.5 | 27.4 | 48.3 |
| MaskRCNN-UltraSAM | point | 75.4 | 98 | 92.5 | 80.1 |
| MFG-UltraSAM | point | 73.8 | 90.1 | 90.1 | 78 |
| MFG-UltraSAM-zeroshot | point | 34.9 | 87.2 | 14.9 | 44.7 |
| MaskRCNN-UltraSAM-zeroshot | point | 41.6 | 91.7 | 28.9 | 52.6 |
| UltraSAM-zeroshot | box | 35.3 | 83.9 | 20.8 | 45.7 |
| MaskRCNN-UltraSAM-zeroshot | box | 28.8 | 87.9 | 25.1 | 50 |

Table 2: Plaque Area Mask Segmentation Quantification Results

| Model | prompt type | mAP | mAP50 | mAP75 | mAR |
|---|---|---|---|---|---|
| UltraSAM | point | 28.5 | 73 | 12.7 | 36.1 |
| UltraSAM-zeroshot | point | 22 | 71.7 | 1.7 | 28.5 |
| MaskRCNN-UltraSAM | point | 4.6 | 22.3 | 2 | 12.2 |
| MFG-UltraSAM | point | 27.4 | 71 | 11.2 | 35.7 |
| MFG-UltraSAM-zeroshot | point | 16.5 | 64.9 | 0.02 | 23.6 |
| MaskRCNN-UltraSAM-zeroshot | point | 10.2 | 36.5 | 1.7 | 15.8 |
| UltraSAM-zeroshot | box | 16.5 | 64.9 | 0.02 | 23.6 |
| MaskRCNN-UltraSAM-zeroshot | box | 8.1 | 30.1 | 1.1 | 45.5 |

## Acknowledgments

This research was supported by the National Institutes of Health grant R01-HL147866. We are grateful to Siemens Medical Solutions USA, Inc., for providing the S2000 Axius Direct Ultrasound Research Interface (URI) and software licenses.

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
