# OpenReview forum: "Segmenting Carotid Plaques with Different SAM Prompts"
_MIDL.io/2025/Short_Papers — MIDL 2025 - Short Papers_

### Official Review · Reviewer_8wyF · 2025-04-16

**Rating:** 5
**Confidence:** 5

**Summary:**

This paper explores the potential of replacing hand-crafted human prompts with model-generated segmentation proposals in Segment Anything style Models (SAM) for application in B-mode ultrasound. Experiments are performed on one dataset using several variants of prompt generation combined with different foundation model flavours. The results showed that automatically generated prompts approach the performance capabilities of ground truth prompting for bounding box detection within ultrasound plaque segmentation.

**Strengths:**

- The methodology of replacing human prompts with machine-generated modality-specific (in this case, ultrasound) ones for querying foundational segmentation models is interesting and technically sound.
- Several model proposals are presented as comparisons and the models seem to approach capabilities of hand-crafted region proposals, which is promising

**Weaknesses:**

- It is not clear what confidence level of IoU the results in table 2 are calculated for.
- No qualitative visualisations are presented, which makes it hard to understand what kind of shortcomings the model has.
- No comparisons against models apart from the SAM family are presented, which does not provide full context for the performance improvements. For example, it would be interesting to know how this method compares against Visual Prompting via Image Inpainting (https://arxiv.org/abs/2209.00647) or nnUNet (https://github.com/MIC-DKFZ/nnUNet). This is perhaps something to consider for future work.
- The computational cost of running inference and training is not specified.

---

### Decision · Program_Chairs · 2025-05-01

Accept